# Fetal Heart Rate < 3rd Percentile for Gestational Age Can Be a Marker of Inherited Arrhythmia Syndromes

**DOI:** 10.3390/jcm12134464

**Published:** 2023-07-03

**Authors:** Nadia Chaudhry-Waterman, Bharat Dara, Emily Bucholz, Camila Londono Obregon, Michelle Grenier, Kristen Snyder, Bettina F. Cuneo

**Affiliations:** 1The Division of Cardiology, Department of Pediatrics, University of Colorado School of Medicine, Denver, CO 80045, USA; 2Presbyterian Hospital, Albuquerque, NM 87106, USA; 3The Colorado Fetal Care Center, Children’s Hospital Colorado, Aurora, CO 80045, USA; 4The Department of Obstetrics, University of Colorado School of Medicine, Denver, CO 80045, USA

**Keywords:** fetal bradycardia, arrhythmia, LQTS, RYR2, HCN4

## Abstract

Background: Repeated fetal heart rates (FHR) < 3rd percentile for gestational age (GA) with 1:1 atrioventricular conduction (sinus bradycardia) can be a marker for long QT syndrome. We hypothesized that other inherited arrhythmia syndromes might present with fetal sinus bradycardia. Methods: We reviewed pregnancies referred with sinus bradycardia to the Colorado Fetal Care Center between 2013 and 2023. FHR/GA data, family history, medication exposure, normalized isovolumic contraction times (n-IVRT), postnatal genetic testing, and ECGs at 4–6 weeks after birth were reviewed. Results: Twenty-nine bradycardic subjects were evaluated by fetal echocardiography. Five were lost to follow-up, one refused genetic testing, and one had negative genetic testing for any inherited arrhythmia. Six had non-genetic causes of fetal bradycardia with normal prenatal n-IVRT and postnatal QTc. Thirteen carried pathogenic variants in RYR2 (*n* = 2), HCN4 (*n* = 2), KCNQ1 (6), and other LQTS genes (*n* = 4). The postnatal QTc was <470 ms in subjects with RYR2, HCN4, and two of those with KCNQ1 mutations, and >470 ms in subjects with CALM 2, KCNH2, SCN5A, and four of those with KCNQ1 mutations. LQTS and RYR2 mutations were associated with prolonged n-IVRT, but HCN4 was not. Two fetuses died in utero with variants of uncertain significance (CACNA1 and KCNE1). Cascade testing uncovered six affected but undiagnosed parents and confirmed familial inheritance in five. Conclusion: In addition to heralding LQTS, repeated FHR < 3rd percentile for GA is a risk factor for other inherited arrhythmia syndromes. These findings suggest that genetic testing should be offered to infants with a history of FHR < 3rd percentile for GA even if the postnatal ECG demonstrates a normal QTc interval.

## 1. Introduction

The conduction system of the human heart is functionally mature by 6 weeks of gestation [1]. Fetal heart rate (FHR) is inversely related to gestational age (GA) and ranges from about 170 beats per minute (bpm) at 8 weeks to 130–140 bpm after 34 weeks [1,2]. Sinus bradycardia, defined as a sustained FHR < 110 beats per minute with 1:1 AV conduction over at least a 10-min period, can occur secondary to increased vagal tone, fetal distress, or maternal medications. However, prior literature has demonstrated that a more subtle sinus bradycardia (i.e., repeated FHRs < 3rd percentile for GA but > 110 bpm) can be the most common rhythm manifestation of long QT syndrome (LQTS), an inherited channelopathy that can result in stillbirth, sudden infant death, or cardiac arrest at all ages [3,4]. Because FHR < 3rd percentile for GA is faster than 110 bpm—the traditional definition of bradycardia—its significance as a harbinger of fetal LQTS is usually unrecognized, overlooked, or attributed to a “low normal baseline FHR”.

Long QT syndrome is an inherited arrhythmia due to pathogenic variants in ion channel genes. KCNQ1, KCNH2, and SCN5A gene defects are the most common etiologies of LQTS and result in abnormal potassium and sodium channel function. Pathogenic variants in HCN4, responsible for the hyperpolarization-activated channel, alter sinus node activity [5]. Subjects with calcium channel genetic variants such as RYR2 and CASQ2 present as catecholaminergic polymorphic ventricular tachycardia [6,7], and abnormal transcription factor genes NKY2.5, GAT4, and TBX5 result in conduction system disturbances, tachycardias, cardiomyopathies, congenital heart disease, and sudden cardiac death [8]. The phenotypes of inherited arrhythmia syndromes have been primarily described in adolescence and adults, with the exception of LQTS variants KCNQ1, KCNH2 and SCN5A, which are well described before birth [3,9,10,11]. Thus, the prenatal phenotype of most inherited arrhythmias has not been described, or their descriptions are limited. As many of these disorders are progressive and respond to primary prevention, it is important to recognize the earliest signs of disease.

The objective of this study was to determine whether fetal sinus bradycardia, defined as FHR < 3rd percentile for gestational age and known to be a marker for fetal long QT syndrome, might also herald other channelopathies and inherited arrhythmia syndrome. If, indeed, sinus bradycardia is more of a general marker for channelopathies, ascertainment of potential subjects would allow earlier primary prevention of both the index fetal case and affected family members.

## 2. Methods

### 2.1. Cohort

We performed a retrospective case series review of pregnant subjects referred to the Colorado Fetal Care Center at the Children’s Hospital of Colorado from obstetrical care providers because of fetal bradycardia. Inclusion criteria consisted of pregnancies with repeated fetal heart rates < 3rd percentile for gestational age, based on the previously published literature [3]. Subjects were included if, in addition to bradycardia, they had a family history of an inherited arrhythmia, if mothers were on medications or had a condition that resulted in bradycardia, or no cause for fetal bradycardia was found in the maternal history. We searched our REDcap fetal cardiology database for pregnancies referred between 2013 (first database entries) and 2023 with a diagnosis of “bradycardia”. Data from the electronic medical record (maternal medical, obstetric, and family histories) and fetal echocardiograms (fetal heart rate and rhythm, normalized isovolumic contraction times (N-IVRT), cardiac structure and function) were reviewed. We also evaluated the postnatal electrocardiograms and neonatal genetic testing results. The gestational age of the first fetal heart rate < 3rd percentile for gestational age was recorded, as was the longest N-IVRT, any other pertinent findings, such as LV non-compaction, structural heart disease and other rhythms, as well as postnatal ECG and genetic testing results, were included in the results.

The study protocol was approved by the Institutional Review board at Children’s Hospital Colorado and the University of Colorado (IRB #23-0304) as secondary retrospective research not requiring informed consent.

### 2.2. Fetal Echocardiograms

Fetal echocardiography with 2-D, M-mode, and spectral Doppler was performed to assess cardiac anatomy and heart rhythm [12]. The normalized isovolumic contraction time, a surrogate for repolarization time [13], was calculated using spectral Doppler—by dividing the isovolumic contraction time (mitral valve closure to aortic valve opening) by the RR interval—by a physician blinded to the diagnosis. We diagnosed “pseudo” 2° AV block if every other atrial beat was conducted to the ventricle with a normal AV interval. We diagnosed ventricular tachycardia if FHR were >180 bpm, the atrial rate was slower than the ventricular rate, and AV dissociation was seen on simultaneous SVC or the innominate vein and aortic spectral Doppler. Premature ventricular contractions were identified as premature ventricular beats seen on simultaneous ventricular and atrial M-mode.

### 2.3. Postnatal Evaluation

All liveborn infants received echocardiograms at birth and 12-lead ECGs 4–6 weeks after birth. Commercial genetic testing using the cardiac arrhythmia and cardiomyopathy panel from CLIA-approved laboratories was performed after birth. If the newborn had a negative family history but tested positive for a pathogenic variant in one of the inherited arrhythmia genes, cascade testing was offered to the family. After genetic testing was complete, we grouped subjects into 3 diagnoses: inherited arrhythmia syndrome; variants of unknown significance; and non-genetic causes for bradycardia. FHR, QTc, and n-IRVT were compared between these groups.

### 2.4. Statistics

Data were reported as means ± SD. No formal statistical testing was conducted due to the small sample size in each group.

## 3. Results

### 3.1. Fetal Cohort (Figure 1 and Table 1)

The outcomes of the 29 fetuses evaluated for sinus bradycardia are shown in Figure 1. Of the 23 subjects with follow-up, 13 had pathogenic variants in genes associated with LQTS, CPVT, or other conduction abnormalities, 8 of whom (57%) had no known family history at the time of diagnosis. Three additional subjects had a VUS (2 were stillborn but were tested before their deaths), one tested negative for an arrhythmia variant, and six had a non-genetic cause of bradycardia (see below). Of the 13 with pathogenic variants, 9 had LQTS variants, 2 had RYR2 variants, and 2 had HCN4 variants. Of the two stillbirths, #12 had severe growth restriction and an abnormal placenta, which most likely were the cause of death. The mother who had the same variant (p. Ala8Val) was asymptomatic. The second stillbirth occurred in a fetus with a de novo VUS in CACNA1H, the gene coding for the low-voltage T-type calcium channel subunit CAV3.2. This gene is expressed in multiple organs including the brain, liver, kidney, and heart, but to or knowledge, has no known cardiac phenotype. It is not known why the nIVRT was prolonged in this subject.

**Table 1 jcm-12-04464-t001:** Fetuses with repeated HR < 3rd% for GA with inherited arrhythmia as the cause for bradycardia.

ID	GA (wks) First Brady	FHR (bpm)	FHR 3rd% for GA (bpm)	Prolonged n-IRVT	Other Fetal Findings	Postnatal QTc (ms)	Dx
1	12	120	140	yes	TdP, AVB	630	KCNH2 (p. Asn 633 Ser)
2	37	125	128	No data	none	489	SCN5A * (p. Leu1501Pro)
4	37	118	128	yes	PVCs	557	CALM 2 * (p. Asp132Gly)
5	24	124	133	No	none	477	KCNQ1 (p. Ala525Thr)
6	35	121	129	yes		457	RyR2 * del exome 3
7	35	113	129	yes	PVCs	375	RyR2 (p. Ser 3984 Pro)
8	22	104	134	No	VSD, LV non-compaction	421	HCN4 * (p. Asp491Asn
9	12	125	140	No	CoA, VSD, non-compaction	453	HCN4 * (p. sp491Asn)
10	35	100	129	yes	None	390	4 VUS: MYH7, MYPN, TTN x2
11	20	129	134	yes	FGR	IUFD	VUS CACNA1H p. Glu285Gly
12	20	69	134	No	FGR, BVH	IUFD	Maternally inherited VUS KCNE1 p. Ala8Val
13	28	101	131	No	None	442	Neg testing nl echo
14	28	118	131	yes	none	543	KCNQ1 p. Val254Met (mother)
15	22	132	134	No data	none	573	KCNQ1 p. Ala344Val (father)
16	27	127	132	yes	none	465	KCNQ1 S566F (mother)
17	37.2	118	125	yes	none	550	KCNQ1 L266P (mother)
18	27	117	134	yes	none	463	KCNQ1 p. Gly168Ar (mother)

* = parent tested positive for pathogenic variant in offspring.

**Figure 1 jcm-12-04464-f001:**
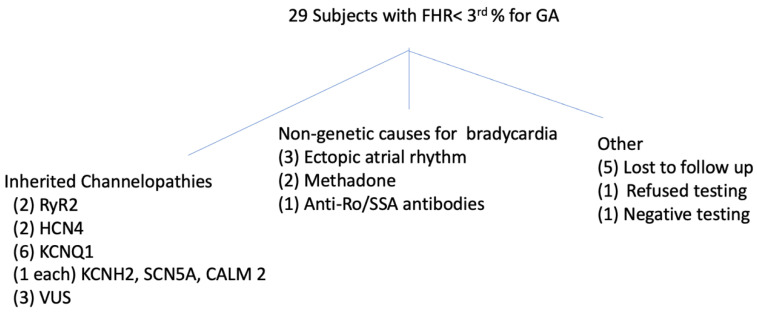
Outcomes of subjects with FHR < 3rd% for GA.

### 3.2. Cascade Testing (Table 1)

Five parents underwent cascade genetic testing following identification of a pathogenic variant in their offspring. The mothers of subjects #2 (SCN5A), #8 (HCN4), and #9 (HCN4) and the fathers of subjects #4 (CALM2) and #6 (RYR2) were found to be positive for the same pathogenic variant as their children. Interestingly, the mothers of #8 and #9 and several family members also had a history of bradycardia but were asymptomatic and never evaluated. The mother of subject #12 carried the VUS found in her fetus with demise. The sister and parents of subject #1 also underwent testing and were found to be negative, both by blood and buccal swab, for the pathogenic KCHN2 variant in their sister/daughter.

### 3.3. Fetuses with Non-Genetic Causes of Sinus Bradycardia (Figure 1 and Table 2)

Six fetuses had a non-genetic cause for sinus bradycardia. In three of these fetuses, cardiac and visceral situs were abnormal and/or systemic venous anomalies were seen by echocardiography. Endocardial fibroelastosis was noted with bradycardia in one fetus with a history of maternal anti-Ro/SSA antibodies. Two fetuses had normal echocardiograms, but their mothers were treated with methadone.

**Table 2 jcm-12-04464-t002:** Fetuses with repeated FHR < 3rd% for GA with non-genetic causes for bradycardia.

ID	GA (wks)	FHR (bpm)	3rd% for GA (bpm)	Prolonged n-IVRT?	Other Findings	Post-Natal QTc (ms)	Reason for Bradycardia
19	33	103	130	No	EFE	402	Anti-Ro antibody
20	30	125	130	No	None	500	methadone
21	38	105	127	no	None	510	methadone
22	34	115	129	No	Systemic venous anomalies	419	Ectopic atrial rhythm LAI
23	24	124	133	No	SVC to LA	438	Absent right SVC
24	22	101	131	No	Systemic venous anomalies	442	Ectopic atrial rhythm, LAI

Key for Table 1 and Table 2: BVH = biventricular hypertrophy; Dx = diagnosis; EFE = endocardial fibroelastosis; FHR = fetal heart rate; FRG = fetal growth restriction; GA = gestational age; IUFD = intrauterine fetal demise; LAI = left atrial isomerism; n-IRVT = normalized isovolumic relaxation time; NSR = normal sinus rhythm; SVC = superior vena cava; TdP = torsades de pointes; VSD = ventricular septal defect; VUS = variant of uncertain significance.

### 3.4. Relationship between Diagnosis, Fetal Heart Rates, QTc and Other Findings (Table 3)

Overall, FHRs were lowest (101–124) in subjects with HCN4 pathogenic variants and those with non-genetic causes of sinus bradycardia. QTc was prolonged (>470 ms) in those with LQTS pathogenic variants, several of whom manifested other arrhythmias, including ventricular ectopy and ventricular tachycardia, and transient in the newborns exposed to methadone. A prolonged IVRT was noted in most (66%) of the LQTS subjects and both RYR2 subjects but not in subjects with HCN4 mutations or non-genetic reasons for bradycardia. In addition to sinus bradycardia, HCN4 fetal subjects had structural cardiac defects (ventricular septal defects in both and an aortic coarctation in one) and non-compaction cardiomyopathy.

**Table 3 jcm-12-04464-t003:** Fetal echo and postnatal ECG findings suggesting diagnosis.

Dx	FHR Mean ± SD (bpm)	n-IVRT Prolonged?	Other Findings	QTc Mean ± SD (ms)
LQTS	121.8 ± 3.3	yes	PVC, 2° AVB, TdP	538.3 ± 71.0
RYR2	117 ± 5.7	yes	PVC	416 ± 58.0
HCN4	114.5 ± 14.9	yes	Non-compaction, CHD	437.0 ± 22.6
Heterotaxy, EAP, anti-Ro antibodies	110.8 ± 10.8	yes	Syst. venous anomalies EFE	425.3 ± 18.5

Key: AVB = atrioventricular block; BPM = beats per minute; EAP = ectopic atrial pacemaker; CL = cycle length; CHD = congenital heart disease; Dx = diagnosis; EFE = endocardial fibroelastosis; FHR = fetal heart rate; IVRT = isovolumic contraction time, ms = milliseconds; LQTS = long QT syndrome; PVC = premature ventricular contractions; TdP = torsades de pointes.

## 4. Discussion

We evaluated the etiologies responsible for repeated FHRs < 3rd percentile for GA and identified several important findings. First, we confirmed that the FHR of LQTS subjects is often higher than the traditional definition of fetal bradycardia; that is, an FHR < 3rd percentile is a more sensitive marker for fetal LQTS than an FHR < 110 bpm. Second, FHR < 3rd percentile can also be found in other inherited arrhythmias and may be the first manifestation of progressive and serious disease. Third, drug-induced QTc prolongation and bradycardia can also affect the fetus.

The relationship between fetal rhythm phenotype and LQTS genotype has been previously defined [3,4,5,6]. Prior studies have suggested that FHRs < 3rd percentile tend to be the most common marker for KCNQ1 and familial KCNH2 pathogenic variants, whereas subjects with certain de novo SCN5A and KCNH2 variants tend to present with more severe bradycardia and ventricular arrhythmias. These subtleties were observed in our study. Fewer data have been reported on the fetal CALM2 genotype, which is causative in some cases of neonatal sudden death [12]. In a recent case report, CALM2 was associated with non-compaction cardiomyopathy [13], although we did not note these findings in the subjects described in this report.

The most noteworthy finding in our study was that that FHR < 3rd percentile for gestational age can be associated not only with LQTS but with also the RYR2 and HCN4 pathogenic variants listed in this report. Abnormalities in the RYR2 gene, which codes for the calcium-gated cardiac ryanodine receptor channel located on the sarcoplasmic reticulum, are associated with catecholaminergic polymorphic ventricular tachycardia (CPVT), a rare congenital arrhythmogenic disorder induced by physical or emotional stress [14]. Unless family history is known or suspected, subjects with CPVT often go undiagnosed until a lethal or near lethal event occurs. To date, there are no reports describing the rhythm phenotype of the fetal RYR2 genotype. Our findings are important because these fetuses often have normal QTc by ECG as neonates and can be mistakenly categorized as “false-positive LQTS subjects” if identified as having sinus bradycardia during fetal life. Because of the overlapping phenotypes of inherited channelopathies causing ventricular tachycardias, other channelopathies such as Brugada syndrome will also present with a normal QTc on a postnatal ECG. Currently, the prenatal findings of Brugada syndrome, if any, are not known, but this diagnosis should be considered if the postnatal ECG is normal but the infant has a family history of Brugada syndrome. With earlier detection and cascade testing, fetal subjects and their families can be counseled earlier in life, which may reduce the risk of fatal ventricular arrhythmias in these individuals.

Fetuses with an inherited arrhythmia and the lowest HRs were positive for a genetic variant in HCN4, which has the least risk of sudden death from ventricular arrhythmias but the greatest likelihood of extranodal disease. Individuals with pathogenic variants in HCN4 usually present with sinus bradycardia along with ventricular non-compaction and structural cardiac defects, including ventricular septal defects and outflow tract anomalies [5,15].

Isovolumetric relaxation time (IVRT) has previously been reported to be a marker for long QT syndrome. The degree of prolonged IVRT can be dependent on gestational age, with the most statistically significant cut-off being those fetuses that are <20 weeks GA [16]. In our study, 66% of those neonates found to have a pathogenic variant for LQTS had prolonged IVRT. This number may not have been 100% because of the gestational ages at which the fetuses were assessed or due to measurement variability between diagnosing physicians. There is also no reported evidence on whether prolonged IVRT in fetuses with inherited LQTS is linked or related to the specific gene mutation; our small cohort would suggest that it is not, as fetuses with KCNQ1, KCNH2, and CALM2 all had prolonged IVRTs.

On the other hand, the findings of subject #5, who had normal nLVIRT but a prolonged postnatal QTc, can be explained based on the results by Clur et al.; in the 21–30 week age group, the N-IVRT was better at excluding disease in normal subjects (specificity of 95% (93–99%) than in identifying disease in the affected, such as subject #5 (sensitivity 47% (24–71%) [16]. Subject #10 had a prolonged nIVRT, but no prolonged QTC. This may have been due to the potential myopathy imparted by the multiple VUSs in several cardiomyopathy genes resulting in subtle cardiac dysfunction.

IVRT was also prolonged in the two fetuses with RYR2 mutations. While this has not previously been reported in the literature, prolonged IVRT, a measure of diastolic dysfunction, is associated with fetal cardiomyopathy and has been reported in adults with CPVT [17,18].

While the most noteworthy CPVT phenotype is exercise or emotion-induced ventricular arrhythmias, including bi-directional ventricular tachycardia, sinus bradycardia has also been reported in 5–50% of subjects [19]. In fact, the findings of sinus bradycardia with exercise-induced VT can be seen even before provocative testing confirms the CPVT diagnosis [20]. Sinus node dysfunction has been elicited by electrophysiological testing in some subjects with RYR2 pathogenic variants [21]. While the basis for fetal bradycardia may be calcium dysregulation, we speculate this bradycardia may be more prominent or noticeable due to increased vagal tone.

Two patients in our cohort were found to have bradycardia in the setting of fetal methadone exposure. The effect of methadone on fetal heart rate and variability has been previously reported and shown to be dose dependent. Higher doses of methadone are associated with lower fetal heart rate and decreased fetal heart rate variability [22]. Additionally, maternal methadone therapy has been shown to result in prolonged fetal QTc intervals, which usually resolve by one week of life [23]. Therefore, for fetuses presenting with bradycardia < 3rd percentile for age or decreased heart rate variability, a thorough social and exposure history of mother and fetus should be conducted to see if methadone exposure may be the cause.

Based on our findings, FHR < 3rd percentile can be a prenatal marker for cardiac pathology. When associated with systemic venous anomalies, an ectopic atrial rhythm is likely. If endocardial fibroelastosis is seen, the mother should be tested for anti-Ro/SSA antibodies. The presence of ventricular ectopy or non-compaction cardiomyopathy suggests an inherited arrhythmia syndrome which can be confirmed by postnatal genetic testing.

## 5. Limitations

There are several limitations of this study; due to the preliminary findings of a retrospective study with a limited cohort, it is difficult to generalize our findings to other inherited arrhythmias, even those within the same category (such as ventricular tachycardia due to other pathogenic variants influencing the calcium channel function such as Casq2 or Trdn) or those with other inherited channelopathies. The reproducibility of our results is in question because the diseases are rare and the sample sizes are small. Additional descriptions of fetuses with pathogenic variants will be necessary if our results are to be generalizable. Other limitations include a referral bias, as not all bradycardic subjects were evaluated, but only those coming to the attention of the referring obstetrical providers. In addition, prolonged N-IVRT has been reported with other conditions, including cardiomyopathy and cardiac dysfunction, which, rather than repolarization abnormalities, may be the primary influences on our N-IVRT findings. Lastly, five subjects with bradycardia were lost to follow-up and may have had normal ECGs and benign genetic testing, which will have influenced our conclusions.

## 6. Conclusions

In this study on a small cohort of fetuses with inherited arrhythmia syndromes, FHR < the 3rd percentile for GA was a marker not only for those with LQTS but also for those carrying RYR2 and HCN4 pathogenic variants. Clinicians should consider postnatal genetic testing in subjects with repeated FHRs < 3rd percentile for gestational age, even in the absence of a prolonged QTc. If genetic testing is positive in the infant, cascade testing can lead to the ascertainment and primary treatment of affected relatives.

## Data Availability

All data is available as part of this manuscript. There is no additional data that is not included in the tables published within the text of the manuscript.

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
