# Peer review of "Fetal Heart Rate < 3rd Percentile for Gestational Age Can Be a Marker of Inherited Arrhythmia Syndromes"

_jcm, 2023, doi:10.3390/jcm12134464_

Round 1

Reviewer 1 Report

Thank you very much for the opportunity to review this paper.

In this retrospective study the authors aimed to evaluate to what extent fetal sinus bradycardia (not the “obstetric definition”, rather sinus bradycardia < 3rd centile) is indicative of inherited arrhythmia syndromes, and not only of LQTS, where this association is well known.

The introduction provides sufficient background to the topic.

Methods are well described and the results are clearly presented and discussed; the conclusions are supported by the results. Naturally, the case numbers are low here, but this was stated by the authors in the Limitations.

In several fetuses (see Table 1) n-LV IVRT was prolonged, but ECG was normal postnatally. Please comment on this. What are possible reasons? Please add this to the manuscript.

And also, ID 5 had a normal n-LV IVRT, but QTc interval was prolonged in the ECG – please comment on this. Please add this to the manuscript.

ID 10 showed prolonged n-LV IVRT, but had a normal ECG postnatally, please comment on this, and also on the found VUS in the CM genes (and TTN – myopathy?). Please add this to the manuscript.

ID 12 died in utero and a VUS was found in KCNE1 (maternally inherited): in absence of any other causes for IUD: would you label this variant as pathological, and not longer as VUS? Please add this discussion also to the manuscript.

ID 11 also died in utero, had a prolonged n-LV IVRT, and had a VUS in CACNA1H gene – please add a comment on this variant to this manuscript, in particular what diseases can be associated with CACNA1H gene variants. Is there a known cause for IUD here? Why would the n-LV IVRT be prolonged here – are there known associated cardiac phenotypes?

Please discuss possible reasons why fetal CPVT is associated with fetal sinus bradycardia and add this discussion to the manuscript.

In summary this article presents important data on the relationship of fetal sinus bradycardia and inherited arrhythmia syndromes – not only LQTS! - which should help not to overlook a repeated subtle fetal sinus bradycardia, prompt the sonographer to measure e.g. nLV-IVRT, and discuss further prenatal (amniocentesis) or postnatal diagnostic procedures such as ECG, genetic testing, family screening etc. It is indeed very interesting that fetal sinus bradycardia is also associated with RYR2 genetic variants (usually we think of CPVT in this context) or HCN4 variants, with a broad variety of phenotypes such as BrS (I recommend to add this association to the manuscript as well), NCCM or sick sinus syndrome. I fully agree that diagnosing early CPVT is extremely important as this disease is often undiagnosed and lethal during childhood.

With some minor amendments (please see above) I fully support the publication of this manuscript in the JCM.

Please see above. 

Author Response

Editor, Journal of Clinical Medicine

June 15, 2023

Dear Sir or Madam,

Thank you and the other reviewers for your review of our manuscript, “Fetal heart rate < 3rd percentile can be a marker of inherited arrhythmia syndromes”. We appreciate the opportunity to respond to reviewer #1’s comments.

Of note, the lines referenced below are based on the final revised version, not the “tracked changes” version.

  1. In several fetuses (Table 1) n-IVRT was prolonged, but postnatal ECG was normal. Please comment on this. What are the possible reasons? Please add this to the manuscript.

We appreciate the reviewer’s comments.

The two fetuses in table 1 of the manuscript both had RYR2 pathogenic variants. A prolonged n-IVRT was previously reported (Clur, 2018) in fetuses with long QT syndrome, but not in any other channelopathies. While the IVRT is a surrogate for repolarization, it also encompasses an important element of cardiac function, and can be prolonged in intrauterine growth restriction (Patey, 2019), maternal asthma (Laleli, 2023), cardiomyopathies (Pedra, 2002).Since pathogenic variants in the RYR2 gene have also been associated with cardiomyopathy, we speculate that with this particular genetic variant, cardiac function, rather than repolarization, may be affected.

In order to relay this information in our manuscript we have done the following:

We have added the following paragraph to the discussion (lines 208 – 217), addressing the discrepancies between the n-IVRT and QTc results.

“Isovolumetric relaxation time (IVRT) has previously been reported to be a marker for long QT syndrome. The degree of prolonged IVRT can be dependent on gestational age, with the most statistically significant cut-off being those fetuses that are < 20 weeks GA. 17 In our study 66% of those neonates found to have a pathogenic variant for LQTS has prolonged IVRT. This number may not have been 100% because of the gestational ages at which the fetuses were assessed or due to measurement variability between diagnosing physicians. There is also no reported evidence on whether prolonged IVRT in fetuses with inherited LQTS is linked or related to the specific gene mutation. Our small cohort would suggest that it is not, as fetuses with KCNQ1, KCNH2, and CALM2 all had prolonged IVRTs.”

We have also added the following to the discussion section (lines 225 – 228) to address IVRT in those with RYR2 mutations.  

“IVRT was also prolonged in the two fetuses with RYR2 mutations. While this has not previously been reported in the literature, prolonged IVRT, a measure of diastolic dysfunction is associated with fetal cardiomyopathy and has been reported in adults with CPVT.18, 19

We hope this explanation and additions to the text are acceptable to the reviewer.

  1. ID #5 had a normal n-LVIRT but the QTc level was prolonged. Please comment on this and add it to the manuscript.

We thank the reviewer for the question. We have added the following to the discussion (line 218 – 224):

“On the other hand, the findings of subject #5, who had normal nLVIRT but a prolonged postnatal QTc, can be explained based on the results by Clur et. Al. in the 21-30 week age group the N-IVRT was better at excluding disease in normal subjects (specificity of 95% (93-99%) than in identifying disease in affectids, such as subject #5 (sensitivity 47% (24-71%).17 Subject #10 had a prolonged nIVRT but the QTC was not. This may have been due to the potential myopathy imparted by the multiple VUSs in several cardiomyopathy genes resulting in subtle cardiac dysfunction.”

We hope this explanation meets with the reviewer’s approval.

  1. ID#10 showed prolonged n-IVRT and had a normal ECG postnatally. Please comment on this and the VUS found in the CM genes (and TTN myopathy) and please add it to the manuscript.

Please see answer to #2.

  1. ID #12 died in utero and a maternally inherited VUS was found in KCNE1. In the absence of any other causes for IUFD would you label this variant as pathological and no longer as a VUS? Please add this discussion to the manuscript.

We thank the reviewer for the question. This fetus had many other reasons for an in utero demise: there was biventricular cardiac hypertrophy and most important an abnormal placenta and severe growth restriction. The mother who had the same KCNE1 variant was not deaf, nor did she have a history of syncope or cardiac arrest. Based on these findings and without any other supportive evidence (which we could not find) we would not change the variant classification.

The following has been added to the results in the fetal cohort section (lines 116 – 118), which now reads:

“Of the two stillbirths, #12 had severe growth restriction and an abnormal placenta, which most likely were the cause of death. The mother, who had the same variant (p. Ala8Val), was asymptomatic.”

We also added the variant to Table 1 and apologize for this oversite. We have also corrected the typo for subject 11 which should read “IUFD” but instead reads “IFUD”.

  1. ID #11 also died in utero, had a prolonged N-LVIRT and a VUS in CACNA1H gene. Please add a comment on this variant to the manuscript, in particular what disease can be associated with CACNA1H gene variants. IS there a known cause of IUFD here? Whey would the N-LVIRT be prolonged here….are there any known associated cardiac phenotypes?

The CACNA1H gene codes for one of the low voltage T-type calcium channel subunit CAV3.2. Both loss and gain of function variants have been found in the brain and have caused autism and epilepsy; they also have been associated with mild intellectual disability and global developmental delay (PMID 22985586). The channel coded for by this gene is also found in kidneys, liver, and heart.  Interestingly, an inhibitor of CACNA1H reduces myocardial cell apoptosis after myocardial infarction in a mouse model (PMID 33378039), but I could find no specific cardiac phenotypes associated with CACNA1H variants nor any reference to the CACNA1H gene or its pathogenic variants associated with stillbirth or IUFD. We surmise the fetal loss might be associated with fetal growth restriction.

The following has been added to the results in the fetal cohort section (lines 118 – 122), after the line added in response to question 4. It now reads:

“The second stillbirth occurred in a fetus with a de novo VUS in CACNA1H, the gene coding for the low voltage the T-type calcium channel subunit CAV3.2. This gene is expressed in multiple organs including the brain, liver, kidney, and heart, but to our knowledge, has no known cardiac phenotype. It is not known why the n-IVRT was prolonged in this subject.”

We hope this is satisfactory to the reviewer and thank him/her for the questions.

  1. Please discuss possible reasons why fetal CPVT is associated with sinus bradycardia and add this discussion to the manuscript.

The reviewer poses an excellent question which definitely warrants a response in the discussion.

We have added the following to the discussion (lines 229 – 236):

“While the most noteworthy CPVT phenotype is exercise or emotion-induced ventricular  arrhythmias including bi-directional ventricular tachycardia, sinus bradycardia has also been reported in 5-50% of subjects (PMID 22787013). In fact, the findings of sinus bradycardia with exercise induced VT can be seen even before provocative testing confirms the CPVT diagnosis (PMID 16272262). Sinus node dysfunction has been elicited by electrophysiological testing in some subjects with RYR2 pathogenic variants (PMID 20676970). While the basis for fetal bradycardia may be calcium dysregulation, we speculate this bradycardia may be more prominent or noticeable due to increased vagal tone.”

We appreciate the reviewer’s question which has improved the manuscript (and provided new insights to the authors!)

  1. It is indeed very interesting that fetal sinus bradycardia is also associated with RYR2 genetic variants (usually we think of CPVT in this context) or HCN4 variants with a broad variety of phenotypes such as BRs (I recommend adding this association to the manuscript), NCCM or sick sinus syndrome.

The reviewer raises an important point. For many of the inherited arrhythmia syndromes there is considerable overlap. We have added a sentence (line 194 – 199) which now reads:

“…as having sinus bradycardia during fetal life. Because of the overlapping phenotypes of inherited channelopathies causing ventricular tachycardias, other channelopathies such as Brugada syndrome will also present with a normal QTc on postnatal ECG. Currently, the prenatal findings of Brugada syndrome, if any, are not known, but his diagnosis should be considered if the postnatal ECG is normal, but the infant has a family history of Brugada syndrome.”

We hope this addition is acceptable to the reviewer.

We appreciate the opportunity to respond to the thoughtful and constructive comments from reviewer #1. We hope our manuscript is now suitable for publication in the Journal of Clinical Medicine.

For all the authors who have read the reviews and responses and agree with the resubmission,

Sincerely,

Nadia Chaudry-Waterman, D.O.

Children’s Hospital Colorado

Pediatric Cardiology Fellow

Reviewer 2 Report

1. Lack of Clear Objectives: The article lacks a clear statement of objectives or research questions. It is essential for the authors to clearly state what they aimed to investigate or achieve through their study. Without well-defined objectives, the study's focus and relevance may be unclear.

2. Limited Sample Size: The study is based on a retrospective case series with a small sample size. With only 29 fetuses evaluated, the findings may not represent the broader population. The authors should acknowledge this limitation and discuss its potential impact on the generalizability of the results.

3. Lack of Statistical Analysis: The article mentions that no formal statistical testing was done due to the small sample size. However, statistical analysis is crucial for drawing reliable conclusions from the data. The absence of statistical analysis limits the robustness of the findings and makes it difficult to assess the significance of the reported associations.

4. Incomplete Methodology Description: The article lacks a detailed description of the data collection, analysis, and interpretation methodology. Readers would benefit from a more comprehensive explanation of the study design, inclusion criteria, data collection procedures, and analytical approaches employed. This would help ensure the transparency and reproducibility of the study.

5. Absence of Ethical Considerations: The article does not provide information regarding the participants' ethical approval or informed consent. Ethical considerations are essential in research involving human subjects, and the authors need to address this aspect of their study.

6. Limited Discussion of Limitations: While the authors briefly mention some limitations in the results section, they do not thoroughly discuss or address the potential impact of these limitations on the study's findings. The authors must provide a comprehensive discussion of the limitations and their implications to enhance the interpretation of the results.

Author Response

Editor, Journal of Clinical Medicine

June 15, 2023

Dear Sir or Madam,

Thank you and the other reviewers for your review of our manuscript, “Fetal heart rate < 3rd percentile can be a marker of inherited arrhythmia syndromes”. We appreciate the opportunity to respond to reviewer #1’s comments.

Of note, the lines referenced below are based on the final revised version, not the “tracked changes” version.

Reviewer #2

  1. Lack of clear objectives. The article lacks a clear statement of objectives or research questions.

We apologize for the lack of clarity around the objectives. We have clarified the objectives by changing the text to read as follows (lines 58 – 63):            

“The objective of this study was to determine whether fetal sinus bradycardia, defined as FHR <3rd percentile for gestational age and known to be a marker for fetal long QT syndrome, might also herald other channelopathies and inherited arrhythmia syndrome, If indeed sinus bradycardia is more of a general marker for channelopathies, ascertainment of potential subjects would allow earlier primary prevention of both the index fetal case and affected family members.

We hope these changes clarify our objectives and aims. 

  1. Limited sample size. The study is based on a retrospective case series with a small samples size: with only 29 fetuses evaluated, the finding may not represent the boarder populations. The authors should acknowledge this limitation and discuss the potential impact on the generalizability of the results.

We agree with the reviewer’s comment and apologize for the oversite. We have added the following to the limitations section (lines 253-257):

“There are several limitations to the study, due to the preliminary findings of a retrospective study with a limited cohort, which makes it difficult to generalize our findings to other inherited arrhythmias, even those within the same category (such as ventricular tachycardia due to other pathogenic variants influencing calcium channel function such as Casq2 or Trdn) or those with other inherited channelopathies.”

We hope this addition to the text is sufficient.

  1. 3. Lack of statistical analysis. The article mentions that no formal statistical testing was done due to the small sample size. However, statistical analysis is crucial for drawing reliable conclusions from the data. The absence of statistical analysis limits the robustness of the findings and makes it difficult to assess the significance of the reported associations.

We appreciate the request to add inferential statistics (e.g., p-values) to the paper. However, we are concerned that adding these summaries could be misleading for a few reasons. First, we initially set out to conduct this project as one with descriptive statistics to describe our cohort's experience. Second, because we did not set out to conduct an inferential study, we did not conduct a power calculation to estimate the sample size(s) needed for each group to find a significant effect. Because of this, we suspect any analyses would be underpowered and that could lead to misinterpretations of the findings as not meaningful or important even in the presence of described clinical differences in the descriptive statistics. For example, if we were to focus on specific mutation classes, we only have 2 cases with the HCN4 and 2 cases with RYR2 pathogenic variants. If we were to more coarsely group our observations (e.g., inherited arrhythmias vs non-genetic), it may result in similar but heterogeneous comparisons that could also mislead a reader on the finding. For these reasons, we would feel more comfortable presenting descriptive statistics to avoid the challenges inherent in adding p-values.

  1. Incomplete Methodology description…readers would benefit from a more comprehensive explanation of study design, inclusion criteria, data collection procedures and analytical approaches. This would help to ensure the transparency and reproducibility of the study.

We apologize for the incomplete description of methods and thank the reviewer for the comment. We have completely changed the methods section to the following (line 66 – 81):

“We performed a retrospective case series review of pregnant subjects referred to the Colorado Fetal Care Center at the Children’s Hospital of Colorado from obstetrical care providers because of fetal bradycardia. Inclusion criteria consisted of pregnancies with repeated fetal heart rates < 3rd percentile for gestational age, based on previously published literature (Mitchell). Subjects were included if, in addition to bradycardia, they had a family history of an inherited arrhythmia, if mothers were on medications or had a condition that result in bradycardia, or no cause for fetal bradycardia was found in the maternal history. We searched our REDcap fetal cardiology database for pregnancies referred between 2013 (first database entries and 2023 with a diagnosis of “bradycardia”. Data from the electronic medical record (maternal medical, obstetric, and family histories), and fetal echocardiograms (fetal heart rate and rhythm, normalized isovolumic contraction times(N-IVRT), cardiac structure and function) were reviewed. We also evaluated the postnatal electrocardiograms and neonatal genetic testing results. The gestational age of the first fetal heart rate < 3rd percentile for gestational age was recorded, as was the longest N-IVRT, any other pertinent findings such as LV non-compaction, structural heart disease and other rhythms, as well as postnatal ECG and genetic testing results were included in the results.”

We hope these changes fulfill the reviewer’s requirements.

  1. Absence of ethical considerations. The article does not provide information regarding the participant’s ethical approval or informed consent.

We apologize for the incomplete reporting on ethics. We did of course have IRB approval for the study but added details have been included in the text. Lines 82 - 84 now read as follows: 

“The study protocol was approved by the Institutional Review board at Children’s Hospital Colorado and the University of Colorado (IRB #23-0304) as secondary retrospective research not requiring informed consent.”

We hope these additions are satisfactory to the reviewer.

  1. Limited discussion of limitations. While the authors briefly mention some limitations in the results they do not thoroughly discuss or address the potential impact of these limitations and their implications to enhance the interpretation of the results.

We thank the reviewer for the comment and the opportunity to improve the limitations section. In addition to the changes made in response to point #2, we have also added the following (lines 257 – 266):

“The reproducibility of our results is in question because diseases are rare, and the sample sizes are small. Additional descriptions of fetuses with pathogenic variants will be necessary if our results are to be generalizable. Other limitations include a referral bias, as not all bradycardic subjects were evaluated, but only those coming to the attention of the referring obstetrical providers. In addition, prolonged N-IVRT has been reported with other conditions including cardiomyopathy, and cardiac dysfunction, rather than repolarization abnormalities may be the primary influences on our N-IVRT findings. Lastly, 5 subjects with bradycardia were lost to follow-up and may have had normal ECGs and benign genetic testing which influenced our conclusions.”

We hope this addition is acceptable to the reviewer.

We appreciate the opportunity to respond to the thoughtful and constructive comments from reviewer #2. We hope our manuscript is now suitable for publication in the Journal of Clinical Medicine.

For all the authors who have read the reviews and responses and agree with the resubmission,

Sincerely,

Nadia Chaudry-Waterman, D.O.

Children’s Hospital Colorado

Pediatric Cardiology Fellow

Round 2

Reviewer 1 Report

No further comments, accept in present form. 

Only few typos. 

Reviewer 2 Report

The authors have addressed all my comments for this paper.